# The Immune Profile of Major Dysmood Disorder: Proof of Concept and Mechanism Using the Precision Nomothetic Psychiatry Approach

**DOI:** 10.3390/cells11071183

**Published:** 2022-03-31

**Authors:** Michael Maes, Muanpetch Rachayon, Ketsupar Jirakran, Pimpayao Sodsai, Siriwan Klinchanhom, Piotr Gałecki, Atapol Sughondhabirom, Agnieszka Basta-Kaim

**Affiliations:** 1Department of Psychiatry, Faculty of Medicine, Chulalongkorn University and King Chulalongkorn Memorial Hospital, The Thai Red Cross Society, Bangkok 10330, Thailand; muanpetch.mp@gmail.com (M.R.); 6271001030@student.chula.ac.th (K.J.); atapol.s@gmail.com (A.S.); 2IMPACT Strategic Research Center, Deakin University, Geelong, VIC 3220, Australia; 3Department of Psychiatry, Medical University of Plovdiv, 4002 Plovdiv, Bulgaria; 4Maximizing Thai Children’s Developmental Potential Research Unit, Department of Pediatrics, Faculty of Medicine, Chulalongkorn University, Bangkok 10330, Thailand; 5Center of Excellence in Immunology and Immune-Mediated Diseases, Department of Microbiology, Faculty of Medicine, Chulalongkorn University, Bangkok 10330, Thailand; yokpim@gmail.com (P.S.); siriwanklinchanhom@gmail.com (S.K.); 6Department of Microbiology, Division of Immunology, Faculty of Medicine, Chulalongkorn University, Bangkok 10330, Thailand; 7Department of Adult Psychiatry, Medical University of Lodz, 91-229 Lodz, Poland; piotr.galecki@umed.lodz.pl; 8Laboratory of Immunoendocrinology, Department of Experimental Neuroendocrinology, Polish Academy of Sciences, 31-343 Kraków, Poland; basta@if-pan.krakow.pl

**Keywords:** depression, mood disorders, inflammation, neuroimmunomodulation, cytokines, psychiatry

## Abstract

Major depressive disorder and a major depressive episode (MDD/MDE) are characterized by activation of the immune-inflammatory response system (IRS) and the compensatory immune-regulatory system (CIRS). In MDD/MDE, recent precision nomothetic psychiatry studies discovered a new endophenotype class, namely major dysmood disorder (MDMD), a new pathway phenotype, namely reoccurrence of illness (ROI), and a new model of the phenome of depression. The aim of the present study is to examine the association between ROI, the phenome of depression, and MDMD’s features and IRS, CIRS, macrophages (M1), T helper (Th)1, Th2, Th17, T regulatory, and growth factor (GF) profiles. Culture supernatants of unstimulated and stimulated (5 μg/mL of PHA and 25 μg/mL of LPS) diluted whole blood of 30 MDD/MDE patients and 20 controls were assayed for cytokines/GF using the LUMINEX assay. MDMD was characterized by increased M1, Th1, Th2, Th17, Treg, IRS, CIRS, neurotoxicity, and GF profiles. Factor analysis shows that ROI features and immune-GF profiles may be combined into a new pathway phenotype (an extracted latent vector). ROI, lifetime and recent suicidal behaviors, and severity of depression are significantly associated with immunotoxicity and GF profiles. Around 80.0% of the variance in the phenome is predicted by ROI and neurotoxicity or the IRS/CIRS ratio. The molecular pathways underpinning ROI-associated sensitization of immune/growth networks are transmembrane receptor protein kinase-triggered STAT protein phosphorylation, TLR/NF-κB, JAK-STAT, and the main proliferation/survival PI3K/Akt/RAS/MAPK pathway. In conclusion, MDMD’s heightened immune responses are the consequence of ROI-associated sensitization combined with immunostimulatory triggers.

## 1. Introduction

Cytokines have been implicated in a major depressive episode (MDE) in recent reviews and meta-analyses [1,2]. Maes et al. discovered increased levels of cytokine or cytokine receptor production in the culture supernatant of stimulated peripheral blood mononuclear cells (PBMCs) or in the serum of MDE patients in the 1990s, including interleukin (IL)-2, soluble IL-2 receptor (sIL-2R), IL-1, the sIL-1R antagonist (sIL-1RA), IL-6, sIL-6R, and tumor necrosis factor (TNF)-α (review: [3]). These findings were corroborated by the same laboratory’s findings that MDE is associated with an inflammatory or acute-phase response (APR) characterized by elevated levels of positive AP reactants (including haptoglobin) and complement factors, and decreased levels of negative AP reactants (including albumin) [3]. Additionally, early machine learning findings indicated that clinical depression is associated with an increase in the expression of T cell activation markers, including CD25 + (IL-2R) [3]. The cytokine, monocyte-T lymphocyte, and immune-inflammatory response system (IRS) theories of depression were developed in 1995 because of these observations [3]. These early findings in MDE are now well-replicated in various systematic reviews and meta-analyses which include the effects of antidepressant treatments on cytokines and inflammation [4,5,6,7,8,9,10,11,12,13].

According to a recent review [1], mood disorders are associated with activation of the IRS and the compensatory immune regulatory system (CIRS), which includes increased levels of immunoregulatory products that downregulate the IRS and prevent hyperinflammation, such as sIL-1RA, sIL-2R, IL-4, IL-10, and some AP proteins [1]. MDE is characterized by activation of macrophage M1 (IL-1, IL-6, TNF-α), T helper (Th)1 (IL-2, IFN-γ), Th2 (Il-4, IL-5), Th17 (IL-17), and T regulatory (Treg) cells (IL-10) [1]. Nonetheless, the IRS is more active than the CIRS during the acute phases of mood disorders, resulting in a net activation of the immune system. Moreover, growth factors including platelet-derived growth factor (PDGF), vascular endothelial growth factor (VEGF), and fibroblast growth factor (FGF) are also higher in mood disorders than in controls [14,15]. It is critical to note that an immune profile characterized by neurotoxic cytokines (M1, Th1, and Th17) is another hallmark of depression, leading to the conclusion that this profile may result in neuro-affective toxicity, resulting in the affective and cognitive symptoms of depression [1]. The immunological response is intimately linked to redox mechanisms, including increased formation of reactive oxygen and nitrogen species (RONS), diminished antioxidant defenses, and indicators of enhanced nitro-oxidative stress toxicity (OSTOX), all of which are found in depression [16,17,18]. Suicidal behaviors (SB), both recent and lifetime suicidal ideation and attempts, are associated with activated IRS, CIRS, RONS, and OSTOX pathways [19].

The staging of illness (conceptualized as the reoccurrence of episodes and SBs) is significantly associated with IRS, CIRS, and redox pathways. Thus, in major depressive disorder (MDD) and bipolar disorder (BD), the number of prior depressive and manic episodes, as well as SBs, are associated with an increase in plasma IL-1β, sIL-1RA, IL-6, TNF-α, and neopterin, lowered antioxidant enzyme activities, and damage to lipids and proteins [20,21,22,23], while other findings indicate that immune-inflammatory responses are more pronounced in later stages of illness [24,25]. In BD, we detected that the frequency of episodes is inversely related to the proportions of stimulated antigen-specific activated CD3 + CD4 + T cells, and the expression of early activation markers and the transferrin receptor on CD4 + and CD8 + cells, whereas later stages of illness are characterized by decreased frequencies of activated Treg cells [26]. The association between staging and immunological pathways was explained by the fact that pro-inflammatory signals may sensitize the immune system, as well as by abnormalities in CIRS and proliferative responses [22,26].

In this respect, using a new precision nomothetic psychiatry approach, Maes et al. [27,28,29,30] established: (a) A new reoccurrence of illness index (ROI) of MDD/BD, namely a latent vector (LV) extracted from number of depressive and manic episodes and SB as well. (b) A novel model of affective disorders based on antioxidant gene variants, adverse outcome pathways (immune-redox pathways), ROI, and the phenome, which was conceptualized as a LV extracted from depression, anxiety, clinical, SB, disability, and quality of life ratings. (c) A new ROI-redox pathway phenotype, which was conceptualized as a LV extracted from ROI and redox biomarkers, and (d) a new diagnostic class, namely major dysmood disorder (MDMD), characterized by aberrations in immune-redox biomarkers, increased ROI, and phenome scores. Importantly, we found that MDMD cuts across unipolar MDD and BD and is more prominent than the latter diagnoses, indicating that both are phenotypes of the same disorder [26,28,30]. Nevertheless, there are no data on whether MDMD and increased phenome scores are accompanied by aberrations in IRS/CIRS and an increased immuno-neurotoxicity and whether a ROI-immune pathway phenotype may be constructed by linking ROI and cytokines/growth factors.

In previous studies, we used not only serum/plasma but also a 72 h culture supernatant of unstimulated and LPS + PHA-stimulated diluted whole blood to measure cytokine/growth factor production [14,31,32,33]. While the serum levels of some cytokines (IFN-γ, IL-2, IL-4, and IL-5) are difficult to measure, their ex vivo production in stimulated whole blood is easily measurable [31,32,33,34]. Moreover, the latter method adequately reflects the in vivo cytokine production, especially that of cytokines/growth factors, which are produced by different immune profiles [31,32,33,34]. Nevertheless, there are no data on the cytokine and growth factor profiles of MDMD, SBs, and ROI either in unstimulated or stimulated diluted whole blood cultures.

Hence, the aim of the study is to examine: (a) whether a MDMD class may be constructed that is associated with greater (un)stimulated immune (M1, Th1, Th2, Th17, Treg, IRS, CIRS, T cell growth, growth factor, and neurotoxicity) profiles and an IRS/CIRS ratio, (b) whether ROI is associated with the immune profiles and whether a ROI-immune pathway phenotype may be constructed, and (c) whether ROI and associated IRS and neurotoxicity profiles predict the phenome of depression.

## 2. Methods and Participants

### 2.1. Participants

In this study, we included 20 normal controls and 30 MDE patients recruited from the outpatient clinic of the King Chulalongkorn Memorial Hospital’s Department of Psychiatry in Bangkok, Thailand. According to DSM-5 criteria, the patients were diagnosed with MDE and had moderate to severe depression, as measured by the Hamilton Depression Rating Scale (HDRS). We recruited normal volunteers of both sexes, ranging in age from 18 to 65 years from the same catchment area, namely Bangkok, Thailand. The controls were recruited by word of mouth. Both patients and controls were excluded if they: (a) Had neuroinflammatory, neurodegenerative, or neurological disorders such as multiple sclerosis, epilepsy, Alzheimer’s disease, stroke, or Parkinson’s disease. (b) Had (auto)immune diseases such as chronic obstructive pulmonary disease, cancer, psoriasis, type 1 diabetes, asthma, and inflammatory bowel disease. (c) Had inflammatory or allergic reactions three months prior to the study. (d) Were treated with immunomodulatory drugs (lifetime history), including glucocorticoids, (e) were treated with therapeutic doses of omega-3 or antioxidant supplements or anti-inflammatory medication the month prior to the study, or (f) were pregnant or lactating women.

Other DSM-5 axis 1 illnesses such as psycho-organic disorders, schizoaffective disorders, schizophrenia, obsessive compulsive disorder, post-traumatic stress disorder, and drug abuse disorders were excluded as exclusion criteria for depression patients. Healthy participants were excluded if they had a diagnosis of any DSM-5 axis 1 condition or a positive family history of MDD or BD. We statistically adjusted for the possible impacts of the drug state of the patients, namely sertraline (*n* = 18), other antidepressants (*n* = 8, including fluoxetine, venlafaxine, escitalopram, bupropion, and mirtazapine), benzodiazepines (*n* = 22), atypical antipsychotics (*n* = 14), and mood stabilizers (*n* = 4).

Prior to participating in this study, all controls and patients submitted written informed consent. The research adhered to international and Thai ethical standards and privacy legislation. The study was approved by the Institutional Review Board of Chulalongkorn University’s Faculty of Medicine in Bangkok, Thailand (#528/63), in accordance with the International Guidelines for the Protection of Human Subjects as required by the Declaration of Helsinki, The Belmont Report, the CIOMS Guideline, and the International Conference on Harmonization in Good Clinical Practice (ICH-GCP).

### 2.2. Clinical Measurements

A research assistant with expertise in mood disorders performed semi-structured interviews. To assess the severity of depression symptoms, we utilized the HDRS, 17-item version, given by an expert psychiatrist [35]. The Thai state version of the State-Trait Anxiety Assessment (STAI) is a psychological inventory designed to assess the intensity of state anxiety [36]. To evaluate psychiatric axis-1 diagnoses, the Mini-International Neuropsychiatric Interview (M.I.N.I.) was utilized [37]. To compute the ROI, we registered the number of depressive and (hypo)manic episodes and assessed recent and lifetime suicidal behaviors (SB) using the Columbia-Suicide Severity Rating Scale (C-SSRS) lifeline version [38]. Lifetime suicidal behaviors were assessed as C-SSRS item 1 (lifetime suicidal ideation, namely wish to be dead), and item C-SSRS suicidal behavior, number of actual attempts. Recent SB was conceptualized as a PC (labeled “PC recent SB”) extracted from nine C-SSRS items, namely wish to be dead, non-specific active suicidal thoughts, active suicidal ideation with any methods, active suicidal ideation with some intent to act, active suicidal ideation with specific plan/intent, frequency and duration of suicidal ideation, actual attempts, and total number of actual attempts (all past month). Lifetime SB was conceptualized as a PC (labeled “PC lifetime SB”) extracted from 11 C-SSRS items, namely wish to be dead lifetime, non-specific active suicidal thoughts, active suicidal ideation with any methods, active suicidal ideation with some intent to act, active suicidal ideation with specific plan/intent, frequency and duration of ideation, number of actual attempts, preparatory acts or behavior, and total number of preparatory acts (all lifetime).

### 2.3. Assays

At 8:00 a.m., after an overnight fast (at least 10 h), blood was collected in BD Vacutainer^®^ EDTA (10 mL) tubes (BD Biosciences, Franklin Lakes, NJ, USA). In the present study, we measured cytokines/growth factors in unstimulated and stimulated (PHA and LPS) whole blood culture supernatant [31,32,33]. We used RPMI-1640 medium (Gibco Life Technologies, Carlsbad, CA, USA) supplemented with L-glutamine and phenol red and containing 1% penicillin (Gibco Life Technologies, USA) with or without 5 µg/mL PHA (Merck, Darmstadt, Germany) + 25 µg/mL lipopolysaccharide (unstimulated) (LPS; Merck, Germany). Then, 1.8 mL of each of these two mediums was added to 0.2 mL of whole blood, 1/10 diluted, on 24-well sterile plates. Whole blood was seeded on 24-well culture plates. Each subject’s specimens were separated into unstimulated and stimulated conditions and were incubated for 72 h at 37 °C, 5% CO_2_ in a humidified environment. After incubation, the plates were centrifuged for 8 min at 1500 rpm. Supernatants were carefully removed under sterile circumstances, split into Eppendorf tubes, and promptly frozen at −70 °C until thawed for cytokine/growth factor assays. Appendix A shows the names, acronyms, and official gene symbols of all cytokines/growth factors measured in the current study. Appendix A offers a list of the various immune profiles studied here.

The cytokines/growth factors were quantified using the LUMINEX 200 equipment (BioRad, Carlsbad, CA, USA), a multiplex approach. In summary, supernatants were diluted four-fold with medium and incubated for 30 min with linked magnetic beads. After adding detection antibodies and streptavidin-PE for 30 and 10 min, respectively, the fluorescence intensities (FI) were measured. We chose the (blank analyte subtracted) FI values in the current research for statistical analyses since FI are often a better option than absolute concentrations, particularly when numerous plates are employed [39]. The range of FI values that fall inside the concentration curve is shown in Table 1. All samples of all cytokines were quantifiable, except for IL-7, which displayed an abnormally high number (>30%) of results below the assay’s sensitivity and was, therefore, removed [26]. IL-13 demonstrated an acceptable lower limit of 30% of the sensitivity and therefore could be included in the statistical analysis. The CV values between analyses were less than 11% for all studies.

### 2.4. Statistical Analysis

Analysis of variance (ANOVA) was used to compare scale variables, whereas chi-square tests or the Fisher–Freeman–Halton test were used to compare nominal variables across categories. We used generalized estimating equations (GEE) analysis to investigate the association between MDMD and the immune profiles and cytokines/growth factors. The pre-specified GEE analysis, which used repeated measures (unstructured working correlation matrix, linear scale response, and maximum likelihood estimation as a scale parameter method), included fixed categorical effects of time (unstimulated versus stimulated), groups (MDMD versus simple depression and controls), and time × groups or time-by-continuous variable interactions (e.g., HDRS, STAI, staging, PC lifetime, and recent SV), as well as sex, smoking, age, and BMI as covariates. The primary outcome variables in the GEE analyses were the immune profiles, and if these revealed substantial results, we additionally investigated the individual cytokines/growth factors. Multiple effects of time or group on the immune profiles were corrected using the false discovery rate (FDR) *p*-value [40]. Additionally, we incorporated the drug state of the patients as additional predictors in the GEE analysis to rule out any influence of these potential confounders. The GEE approach allows us to account for significant interactions and confounders while avoiding biased imputations induced by incomplete assessments. Nonetheless, there were no missing values in any of the demographic, clinical, or cytokine/growth factor data analyzed in this investigation (except for IL-7, which was excluded from the analysis). We calculated the estimated marginal means for the groups as well as the time × group interactions and used (protected) pairwise contrasts (least significant difference at *p* = 0.05) to examine differences between groups and time × group interactions. We employed multiple regression analysis (automatic technique with a *p*-to-entry of 0.05 and a *p*-to-remove of 0.06 while evaluating the change in R^2^) to identify the biomarkers that predict the phenome or ROI scores. Multicollinearity was determined using tolerance and VIF and multivariate normality was determined using Cook’s distance and leverage, and homoscedasticity was determined using the White and modified Breusch–Pagan tests. The results of these regression analyses were always bootstrapped using 5.000 bootstrap samples, and the latter are presented if the findings were not concordant. Principal component analysis (PCA) was used to reduce the number of suicidal behavior features and to summarize the information in summary indices or patterns. The first PC was considered to indicate a valid pattern when it explained at least 50.0% of the total variance and all loadings were greater than 0.6. To construct latent vectors (factors) underpinning several indicators, we used exploratory factor analysis (unweighted least squares) with the same quality criteria as described above. In addition, we always estimated the factorability using the Kaiser–Meyer–Olkin test for sampling adequacy (should be >0.6) and Bartlett’s test of sphericity. In addition, we performed partial least squares (PLS)-SEM analysis to derive latent variable (LV) scores only when the factors complied with prespecified quality criteria, namely all factor loadings should be >0.6 at *p* < 0.0001, average variance extracted (AVE) > 0.5, Cronbach alpha > 0.7, composite reliability > 0.8, and rho_A > 0.8 [20]. All statistical analyses (except PLS) were conducted using IBM SPSS windows version 28. Two-tailed tests were used, and statistical significance was set at *p* < 0.05. Using a two-tailed test with a significance threshold of 0.05 and assuming an effect size of 0.23 and a power of 0.80 while considering three groups and intercorrelations of around 0.6, the estimated sample size for a repeated measurement design ANOVA is approximately forty-two.

We developed seed-gene-based protein–protein interaction (PPI) networks based on the differentially expressed proteins (DEPs) in MDMD versus controls. The networks were constructed using STRING version 11.0 (https://string-db.org, accessed on 8 March 2022), a predictive database, and IntAct Molecular Interaction Database (https://www.ebi.ac.uk/intact/, accessed on 8 March 2022), a database based on peer-reviewed articles. We developed zero-order PPIs (consisting entirely of seed proteins), a first-order PPI network (consisting of 50 interactions in the first shell and none in the second shell; set organism: homo sapiens, and a minimum required interaction score of 0.400), and expanded networks, including using OmicsNet (OmicsNet 2.0, OmicsNet, accessed on 8 March 2022). Markov clustering (MCL) analysis was performed using STRING to discover DEP communalities. STRING and the Cytoscape (https://cytoscape.org, accessed on 8 March 2022) plugin Network Analyzer were used to examine the network topology. The network’s backbone was defined as a collection of top hubs (nodes with the greatest degree) and non-hub bottlenecks (nodes with the highest betweenness centrality). STRING was utilized to show the physical interactions between the DEPs, and Gene Ontology (GO) net (GOnet (dice-database.org, accessed on 8 March 2022) to make graphs which contain GO terms and genes. The PPI networks were analyzed for their enrichment scores and annotated terms using the following tools: (a) STRING to establish GO biological processes and molecular functions, diseases, as well as KEGG (https://genome.jp/kegg/, accessed on 8 March 2022) and WIKI (WikiPathways–WikiPathways, accessed on 8 March 2022) pathways, and (b) OmicsNet (using InAct) for establishing REACTOME (European Bioinformatics Institute Pathway Database; https://reactome.org) and PANTHER (www.pantherdb.org/pathway/, accessed on 8 March 2022) pathways, (c) Enrichr (Enrichr (maayanlab.cloud/Enrichr/, accessed on March 8 2022)) to establish Elsevier Pathways visualized using Appyter, and (d) MetaScape (Metascape, accessed on March 2022) to establish molecular complex detection (MCODE) components based on GO terms. The enrichment analysis results are always shown using FDR-corrected *p*-values or q-values.

## 3. Results

### 3.1. Construction of ROI, Phenome Scores, and the MDMD Phenotype

The ROI was computed as the first LV extracted from the number of depressive episodes, total number of episodes, C-SSRS lifetime suicidal ideation, C-SSRS number of lifetime suicidal attempts, and PC lifetime SB (the latter was computed as the first PC extracted from 11 lifetime SB C-SSRS items, and this PC explained 62.21% of the variance while all the items showed loadings > 0.740). The first ROI LV explained 75.6% of the total variance and all loadings were >0.6, namely number of depressive episodes: 0.891, total number of episodes: 0.909, lifetime suicidal ideation: 0.931, number of lifetime suicidal attempts: 0.664, and PC lifetime SB: 923, and showed excellent quality criteria, including Cronbach alpha: 0.916, composite reliability: 0.939, rho_A: 0.939, and AVE: 0.756. In order to compute the phenome score, we (a) computed the PC recent SB by extracting the first PC from the 9 recent SB C-SSRS items, which explains 60.54% of the variance, while all those items showed loadings > 0.6, and (b) computed the first LV extracted from the PC recent SB (0.862), diagnosis (controls: 0, MDE: 1, MDE with psychotic/melancholia features: 2; loading: 0.906), HDRS (0.942), and STAI (0.805). This LV showed excellent quality criteria, including Cronbach alpha: 0.902, composite reliability: 0.932, rho_A: 0.920, and AVE: 0.775. The combined ROI-phenome score was computed as the first LV (explaining 72.5% of the variance) extracted from all five ROI indicators and all four phenome indicators. This LV showed excellent quality criteria with factor loadings that were all higher than 0.646, and Cronbach alpha: 0.951, composite reliability: 0.959, rho_A: 0.957, and AVE: 0.725. We divided the study sample into three non-overlapping groups using the ROI-phenome score and a visual binning procedure (analysis of apparent modes and local minima in the frequency histogram with two cutoff points, namely −0.6 and 0.6). As such, we obtained three study groups, namely controls and depressed patients, divided into these with lower (labeled: simple depression) versus high (labeled: MDMD) ROI-phenome scores.

### 3.2. Demographic and Clinical Data of the Study Groups

Table 1 shows the features of the normal controls, simple depression, and MDMD. There were no significant differences in age, sex, or education between the study groups, while depressed patients showed a somewhat higher BMI than controls. The MDMD group showed a higher prevalence of depression with melancholia/psychotic features, number of depression and all episodes, and PC recent and lifetime SB scores.

### 3.3. Differences in Immune Biomarkers between MDMD and SD

Table 2 shows the measurement of unstimulated and stimulated immune profiles in controls, simple depression, and MDMD. The time x group interactions of all immune profiles were significant, and these differences remained significant after FDR *p*-correction (at *p* < 0.04). The interaction patterns showed that the stimulated production of all those profiles was significantly higher than the unstimulated production (all *p* < 0.001) and that the production was always higher in MDMD than in controls, while there were no differences between simple depression and controls. Moreover, the M1, Th1, T cell growth, growth factor, and neurotoxicity profiles were significantly higher in MDMD than in simple depression. In the GEE analyses, there were no significant effects of age, sex, BMI, and TUD. In addition, no significant effects of any of the drugs could be found on any of the immune profiles or single cytokines/growth factors, even without FDR *p*-correction. For example, there were no significant effects of sertraline (W = 1.01, df = 1, *p* = 0.314), other antidepressants (W = 0.895, *p* = 0.344), benzodiazepines (W = 0.91, *p* = 0.340), mood stabilizers (W = 0.10, *p* = 0.745), and atypical antipsychotics (W = 0.59, *p* = 0.443) on the neurotoxicity index.

Table 3 shows the secondary analyses performed on all separate cytokines/growth factors. We found significant interaction patterns between time x group with significantly increased stimulated production of sIL-1RA, IL-5, CXCL-8, IL-9, IL-15, IL-17, FGF, IFN-γ, CXCL10, PDGF, CCL5, TNF-α, VGEF, and G-CSF in MDMD versus controls, while there were no significant differences between controls and simple depression. Moreover, the interaction terms showed that the stimulated production of sIL-1RA, FGF, PDGF, and IFN-γ was significantly greater in MDMD than in simple depression.

### 3.4. Associations with the Features of MDMD

To examine the associations between the immune profiles and the key features of MDMD, we performed GEE analyses and analyzed the interactions between time × HDRS, time × ROI, and time × PCs lifetime/recent SB. Table 4 shows significant time × ROI interactions for all nine immune profiles, and these effects remained significant after FDR *p*-correction (at *p* < 0.043). GEE analyses performed on the solitary cytokines/growth factors showed significant time × ROI interactions for sIL-1RA (W = 8.18, *p* = 0.004), IL-5 (W =7.08, *p* = 0.008), CXCL8 (W = 6.22, *p* = 0.013), IL-9 (W = 5.95, *p* = 0.017), IL-15 (W = 9.86, *p* = 0.002), IL-17 (W = 4.03, *p* = 0.045), FGF (W = 6.06, *p* = 0.014), G-CSF (W = 7.81, *p* = 0.005), GM-CSF (W = 5.75, *p* = 0.017), IFN-γ (W = 5.60, *p* = 0.018), PDGF (W = 6.58, *p* = 0.010), CCL5 (W = 5.22, *p* = 0.022), TNF-α (W = 5.57, *p* = 0.018), and VGEF (W = 5.52, *p* = 0.019).

Table 4 shows significant time x HDRS interactions for all immune profiles (except Th17 and CIRS), and these differences remained significant after FDR *p*-correction (at *p* < 0.036). GEE analyses performed on the solitary cytokines/growth factors showed significant time × HDRS interaction effects for sIL-1RA (W = 5.40, *p* = 0.020), IL-5 (W = 7.32, *p* = 0.007), CXCL8 (W = 6.78, *p* = 0.009), IL-9 (W = 7.32, *p* = 0.007), IL-12 (W = 4.37, *p* = 0.037), IL-15 (W = 17.73, *p* < 0.001), IL-17 (W = 7.18, *p* = 0.007), G-CSF (W = 8.05, *p* = 0.005), IFN-γ (W = 5.71, *p* = 0.017), PDGF (W = 7.39, *p* = 0.007), CCL5 (W = 5.61, *p* = 0.018), TNF-α (W = 4.08, *p* = 0.043), and VGEF (W = 8.17, *p* = 0.004).

Table 4 shows that all time × PC lifetime SB interactions were significant, except Th17 and Th2, and that the effects remained significant after FDR *p*-correction (at *p* < 0.042). GEE analyses performed on the cytokines/growth factors showed significant time × PC lifetime SB interactions for sIL-1RA (W = 12.34, *p* < 0.001), IL-15 (W = 6.75, *p* = 0.009), FGF (W = 4.55, *p* = 0.039), IFN-γ (W = 4.09, *p* = 0.043), CCL5 (W = 4.50, *p* = 0.039), and VGEF (W = 7.15, *p* = 0.006). In addition, there were significant interactions between time × PC recent SB and M1, Th1, Th2, IRS, T cell growth, and NT profiles. GEE analyses performed on the cytokines/growth factors showed significant time × recent SB interactions for sIL-1RA (W = 8.0, *p* = 0.005), IL-2 (W = 4.30, *p* = 0.038), GM-CSF (W = 5.44, *p* = 0.020), CXCL10 (W = 4.29, *p* = 0.038), CCL3 (W = 4.34), *p* = 0.037), and VGEF (W = 3.97, *p* = 0.046).

### 3.5. Construction of a ROI-Immune Score and Prediction of the Phenome

To delineate the prediction of the phenome using the immune profiles, we performed multiple regression analysis with the phenome score as the dependent variable and the immune profiles as explanatory variables, while allowing for the effects of age, sex, BMI, education, and smoking. The phenome score was associated with the residualized M1 (partial correlation coefficient = 0.438, *p* = 0.002), Th1 (r = 0.432, *p* = 0.002), Th2 (r = 0.404, *p* = 0.005), Th17 (r = 0.382, *p* = 0.008), IRS (r = 0.473, *p* < 0.001), CIRS (r = 0.302, *p* = 0.039), neurotoxicity (r = 0.459, *p* = 0.001), T cell growth (r = 0.477, *p* = 0.001), and growth factor (r = 0.450, *p* = 0.001) values after partialling out the effects of the unstimulated levels. These effects remained significant after FDR *p*-correction. We found that 82.3% of the variance in the phenome score was predicted (F = 23.88, df = 8/41, *p* < 0.001) by ROI (β = 0.747, t = 10.09, *p* < 0.001), neurotoxicity (β = 0.434, t = 3.67, *p* < 0.001), CIRS (β = −0.275, t = −2.31, *p* = 026), and age (β = −0.226, t = −3.16, *p* = 0.003), while education (*p* = 0.234), sex (*p* = 0.095), BMI (*p* = 0.393), and TUD (*p* = 0.540) were all not significant. Figure 1, Figure 2 and Figure 3 show the partial regressions of the phenome on staging, residualized neurotoxicity, and CIRS values, respectively. Moreover, substituting the neurotoxicity and CIRS in this regression with the IRS/CIRS ratio (computed as z IRS—z CIRS) showed that the IRS/CIRS ratio was significantly associated with the phenome score (β = 0.418, t = 2.98, *p* = 0.005). Figure 4 shows the partial regression plot of the phenome score on the z IRS/CIRS ratio.

We found that one general factor could be extracted from the residualized neurotoxicity (loading: 0.645), T cell growth (0.719), growth factor (0.702) scores, number of depressive episodes (0.884), number of all episodes (0.976), and PC lifetime SB (0.777). This factor explained 57.6% of the variance and showed adequate Cronbach alpha (0.894), composite reliability (0.904), and rho_A (0.932) values. This ROI-IMMUNE construct was strongly associated (t = 18.76, *p* < 0.001) with the phenome score, with a zero-order correlation coefficient of 0.825 and a partial correlation coefficient of 0.799, i.e., after adjusting for the effects of age (t = −2.70, *p* = 0.010), sex, BMI, and education (all non-significant). There were no significant effects of the five drug state variables on the phenome scores.

### 3.6. Results of Network, Annotation, and Enrichment Analysis

The protein network of the 14 DEPs that are elevated in MDMD is shown in Figure 5A. It consists of 14 nodes with 79 edges, exceeding the predicted number (*n* = 11), with a *p*-enrichment value of <1.0 × 10^−16^. This network has an average node degree of 11.3 and an average local clustering coefficient of 0.967. Additionally, we built a first-order PPI network with 50 interactions in the first shell and none in the second, and this network has 64 nodes and 760 edges, exceeding the predicted amount (*n* = 177; *p* < 1.0 × 10^−16^). This network has an average node degree of 23.8, a mean local clustering coefficient of 0.707, a network diameter of 3, a radius of 2, a typical path length of 1.651, a network density of 0.377, and a heterogeneity of 0.511. In decreasing order of significance, the top five seed hubs were: TNF (degree = 50), VEGFA (45), CXCL8 (42), IFNG (39), and CSF3 (37). FGF2 (betweenness centrality = 0.0275) and PDGFA were the top two non-hub bottlenecks (0.0147). As a result, the first-order PPI network’s backbone is composed of four immune DEPs and three growth factor DEPs. MCL cluster analysis (inflation parameter: 3) indicated two communalities, one focusing on immunological DEPs and the other on growth factors (see Figure 5B for a first-order network).

Table 5 summarizes the results of an enrichment study performed on all DEPs in MDMD’s first-order PPI network and the KEGG pathway classifications determined using STRING. The most over-represented KEGG pathways were immune-inflammatory pathways and viral infections, as well as JAK-STAT and MAPK pathways. Additionally, this table summarizes the KEGG pathways that were significantly enriched in the second growth factor cluster, including the RAS, MAPK, Rap1, and PI3K-Akt signaling pathways. Additionally, the same table includes the diseases that were enriched in all immunological DEPs associated with MDMD. Figure 6 shows the top ten Elsevier Pathways that are over-represented in the first-order protein–protein interaction network of MDMD.

The results of MCODE analysis for all first-order DEPs utilizing GO biological and molecular terms are shown in Table 6. We identified three molecular complexes that represent cytokine responses, nuclear factor (NF)-B signaling regulation, and necrotic cell death. MCODE analysis of growth factor DEPs revealed the presence of a single cluster of transmembrane receptor protein kinase activity. Table 7 summarizes the top ten REACTOME pathways that were enriched in the first-order PPI network (as evaluated using OmicsNet), mostly involving Toll-Like Receptor signaling. Additionally, the same table illustrates the PANTHER biological processes that are enriched in the PPI network, with a particular emphasis on transcription through RNA polymerase, viral activities, and angiogenesis.

The GOnet enrichment analysis (biological process, q value threshold of <0.0001; *p*-value threshold of <2.5 × 10^−9^) is shown in Figure 7, highlighting the 14 seed MDMD DEPs and the significant GO annotations. The GO terms that are substantially over-represented in the DEP list include inflammation, proliferation, STAT protein phosphorylation, JAK-STAT pathway regulation, and cellular responses to stress, lipids, and LPS.

## 4. Discussion

### 4.1. Activated Immune Profiles in MDMD

The first key conclusion of this research is that an acute episode of MDMD is characterized by enhanced activation of all immune profiles and 14 of the 23 cytokine/growth factors in stimulated (but not unstimulated) diluted whole blood, as compared with controls. Notably, immunological profiles and single cytokines/growth factors did not differ between simple depression and controls, but were consistently greater in MDMD than in simple depression, with M1, Th1, T cell growth, and neurotoxicity profiles, sIL-1RA, FGF, and IFN-γ being significantly higher. These results support the IRS/CIRS hypothesis of depression, which states that in depression, IRS profiles such as M1, Th1, Th17, as well as CIRS profiles, particularly Treg and Th2, are considerably active [1]. We reviewed the many previous data on the serum cytokines when MDD/MDE was compared to controls [1,3,14]. Nonetheless, most of these findings are not entirely comparable to ours since we focused on MDMD, a subgroup of MDD/MDE. One significant contrast with past findings concerns the Th1 profile, as a meta-analysis [2] revealed decreased IFN-γ production in depression, but the present and earlier investigations found enhanced stimulated IFN-γ production [41]. Nonetheless, serum IFN-γ levels are often undetectable, and hence it is contradictory to assume that depressed individuals have lower blood levels of this cytokine than the barely detectable control values. Additionally, increased IFN-γ production was identified as a characteristic of MDMD in the present investigation, highlighting the critical involvement of the Th1 phenotype and cell-mediated immunity in that illness [42].

Notably, the current research revealed that MDMD was associated with a significantly elevated neurotoxicity profile, supporting the neurotoxicity theory of depression [1]. MDMD exhibits increased production of IL-15, IL-17, TNF-α, IFN-γ, CXCL8, CXCL10, and CCL5, all of which have neurotoxic properties [1,43]. Notably, the adverse effects of those cytokines may exacerbate the neurotoxic consequences of increased RONS, lipid and protein oxidation, aldehyde formation, hypernitrosylation, and decreased antioxidant and neurotrophic defenses in MDMD [28,29]. Additionally, this research showed that MDMD is associated with elevated levels of the growth factors VEGF, PGDF, and FGF. Previously, a meta-analysis revealed elevated FDF levels in depression, however results on VEGF and PDGF levels were more controversial [15,44,45,46,47,48]. Nonetheless, the present study’s findings are more appropriate since stimulated cultures of diluted whole blood more accurately mirror the in vivo scenario, and MDMD is a more accurate model than MDD/MDE. Interestingly, these three growth factors regulate cell division, endothelial cell chemotaxis, mitogen-activated protein kinase (MAPK) signaling pathways, and angiogenesis (3 items (human)—STRING interaction network (string-db.org, accessed on 8 March 2022)), and as such these growth factors may contribute to the increased immune responses.

Our results confirm that the clinical study population for major depression is divided into two clinically and biologically different subtypes, i.e., MDMD and simple depression. We generated the diagnosis of MDMD in this investigation using previously described clinical characteristics [28,29]. As such, immune profile tests were used as a proof of concept of the MDMD construct. The primary goal of precision psychiatry should be to define the correct (reliable and cross-validated) model of depression as a major psychosis or a severe medical condition and to distinguish this class from depressive-like emotional reactions [30]. Thus, MDMD is clinically defined by an increase in ROI and phenome scores, as well as an increase in melancholia and psychotic symptoms, suicidal behaviors, and disabilities, lowered quality of life, as well as activated IRS/CIRS/neurotoxin/growth factor and redox pathways (this study and [28,29,30]). As a result, future research on depression should always take MDMD into account rather than MDD or MDE [30,49].

### 4.2. Immune Profiles and the Features of MDMD

The second main conclusion of this research is that activated immune profiles are strongly linked with MDMD characteristics, including increased suicidal behaviors, staging, and severity of the phenome. To begin, both lifetime and current suicide behaviors are related with increases in most immunological profiles, most notably the IRS and neurotoxicity profiles. Again, in light of the IRS/neurotoxicity hypotheses of suicidal ideation and attempts [19], the neurotoxic profile and neurotoxic cytokines and chemokines (e.g., IL-2, IL-15, CCL5, and CXCL10) may exacerbate the adverse effects of RONS/OSTOX on suicidal behaviors [19].

Secondly, by integrating the number of depressive and total episodes and lifetime suicidal behaviors into a ROI or staging index, we were able to corroborate the ROI model [20,28,30]. Furthermore, our findings revealed that ROI and activated immune-redox pathways are not only intrinsically linked but also generate a novel ROI-immune pathway phenotype, indicating that ROI and IRS, neurotoxicity, and T cell growth profiles are manifestations of a common core in MDMD. These results corroborate our prior findings that ROI and immune-redox indicators belong to a common core [28,30]. As discussed in the introduction, there is some evidence that the frequency of depressive and manic episodes is related to serum IRS (sIL-1RA, IL-6) and CIRS (IL-10) biomarkers and to the stimulated expression of activation markers [26]. It is worth comparing this to “epileptogenic basolateral amygdala kindling,” a model for the recurrence of epileptic seizures in which a seizure raises the probability of subsequent seizures [50]. Notably, a shared core underpins the frequency and uncontrollability of seizures, comorbid depression and anxiety, and redox pathways, showing that this pathway phenotype identifies a particularly severe form of TLE [30]. Similarly, the kindling theory of affective disorders states that as episodes recur, the latter become increasingly sensitized [51], and our findings indicate that this “affective kindling” is a central feature of MDMD and is caused by aberrations in immune-redox pathways [28,30].

Thirdly, the MDMD phenome might be understood as a common core comprised of the intensity of depression and anxiety, the presence of melancholia or psychotic features, and recent suicidal behaviors which incorporate various aspects of suicidal ideation and attempts. Most notably, a considerable proportion (around 80%) of the variance in the phenome score may be attributed to ROI and the IRS or neurotoxicity profiles (positively related) and CIRS and age (inversely associated). These results corroborate the IRS/CIRS hypothesis of affective disorders [1], demonstrating that the neurotoxicity profile may result in affective disorders, particularly in people with weakened or inadequate CIRS defenses. Additionally, the results corroborate our prior findings that the ROI-redox pathway phenotype substantially predicts the phenome of mood disorders [28,29,30].

### 4.3. Increased Immune Responsivity in MDMD

The third important conclusion of this research is that MDMD and its features relate to the stimulated immune responses and not the unstimulated activity levels of the immune profiles. As such, the stimulated immune/growth factor production is part of the ROI and may, therefore, be ascribed to sensitization (kindling) processes [20,21,22,23,26]. Since ROI had no effect on unstimulated profiles, we may infer that not only ROI-induced sensitization but also continuous immunostimulatory triggers are required to account for the enhanced immune and growth factor responses. By inference, because our ex vivo stimulated whole blood assay reflects the measurement of serum cytokines in vivo, we can deduce that the elevated serum cytokine levels frequently observed in depression [1,2] are a result of both sensitization and the continuous presence of immunostimulatory triggers during the acute phase of illness. Recently, a variety of trigger factors have been identified that may account for these effects, including increased LPS load from increased bacterial translocation and leaky gut [52], LPS particles on the membrane of outer membrane vesicles that circulate in the serum [53], increased LPS contents due to apical periodontitis [54], continuous stimulation of the Toll-Like Receptor (TLR)-Redox cycle by damage-associated molecular patterns, including oxidatively modified neoepitopes which are abundant in affective disorders [55], other pathogen-associated molecular patterns, including due to latent viral infection such as with cytomegalovirus [26], and immunostimulatory effects of psychological stressors and medical disorders that activate the cytokine network [56]. The impact of the latter was further corroborated by our disease enrichment analysis, which showed that a multitude of (auto)immune diseases are enriched in the PPI network built using the DEPs of MDMD (see listings in Figure 6 and Table 5).

Notably, we observed no impact of antidepressant intake on the ex vivo (un)stimulated immunological profiles, even though these medications have a strong negative immunoregulatory effect in vitro, attenuating cell-mediated immune responses by lowering IFN-γ production while raising IL-10 production [31,32]. By extrapolation, despite antidepressants’ inherent immunoregulatory properties, IRS and neurotoxicity profiles in serum and stimulated whole blood cultures remain elevated, because of the continual stimulation throughout the acute phase of MDMD. By inference, these trigger factors as well as the molecular networks that underpin the sensitized immune-growth responses are new drug targets to treat affective disorders.

Using annotation and enrichment analysis, we were able to pinpoint the most important molecular pathways and functions that play a role in MDMD, namely crosstalk between transmembrane receptor protein kinase-activated STAT protein phosphorylation, hyper-responsive TLR/NF-κB and JAK-STAT pathways [56], and hyper-responsivity of the main proliferation/survival pathway, namely PI3K/Akt/RAS/MAPK signaling [57].

## 5. Conclusions

Increased M1, Th1, Th2, Th17, Treg, IRS, CIRS, neurotoxicity, and growth factor profiles are seen in MDMD. ROI, suicidal behaviors, and the phenome of depression are all linked to immunotoxicity and growth factor profiles. Factor analysis revealed that ROI characteristics and immune-growth factor profiles may be used to construct a novel pathway phenotype. ROI and immune profiles accounted for around 80.0 percent of the variance in the phenome. The exaggerated immune responses in MDMD may be explained by ROI-induced sensitization coupled with continuous immunostimulatory triggers. This study delineated new drug targets to treat affective disorders, namely ROI-associated sensitization of immune and growth factors and the underlying molecular pathways, that is the transmembrane receptor protein kinase-triggered STAT protein phosphorylation, TLR/NF-B, JAK-STAT, and the proliferation/survival PI3K/Akt/RAS/MAPK pathway.

## Figures and Tables

**Figure 1 cells-11-01183-f001:**
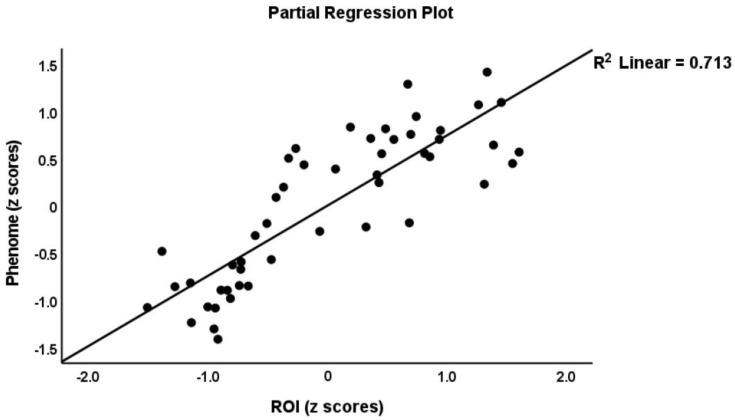
Partial regression of the phenome score on the recurrence of illness (ROI).

**Figure 2 cells-11-01183-f002:**
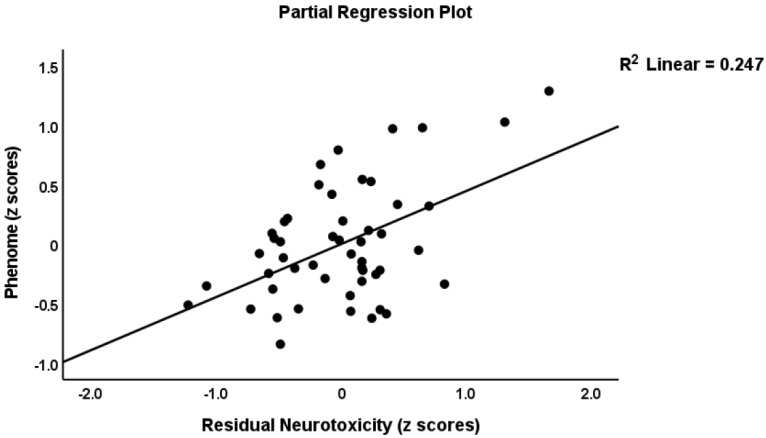
Partial regression of the phenome score on the immune-neurotoxicity index.

**Figure 3 cells-11-01183-f003:**
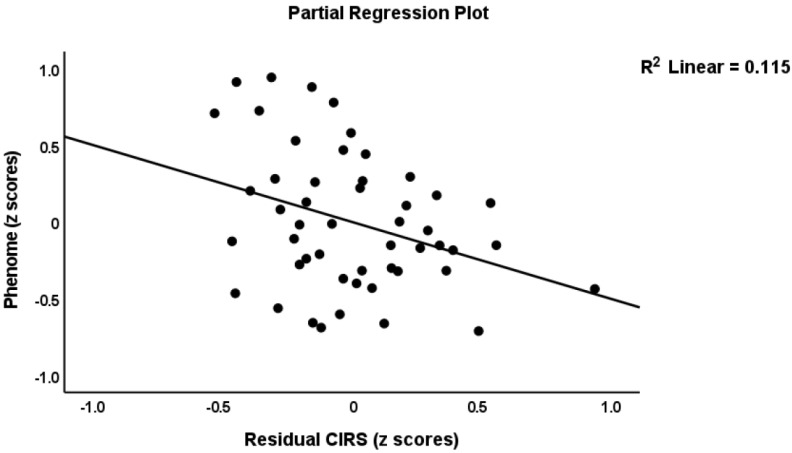
Partial regression of the phenome score on an index of the compensatory immune-regulatory system.

**Figure 4 cells-11-01183-f004:**
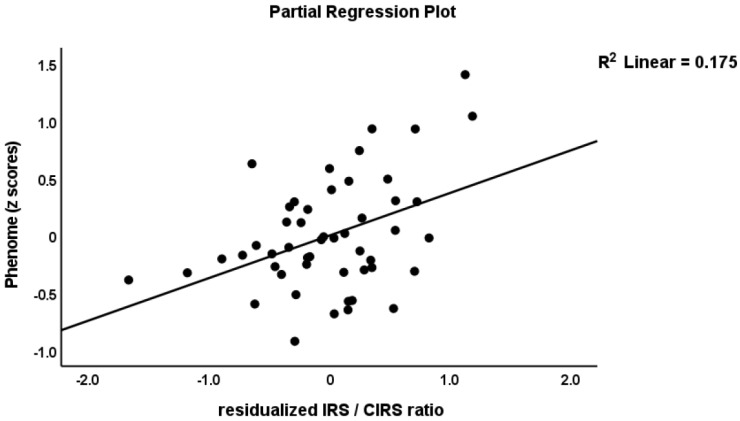
Partial regression plot of the phenome score on the immune-inflammatory response system (IRS)/compensatory immune-regulatory system (CIRS) ratio.

**Figure 5 cells-11-01183-f005:**
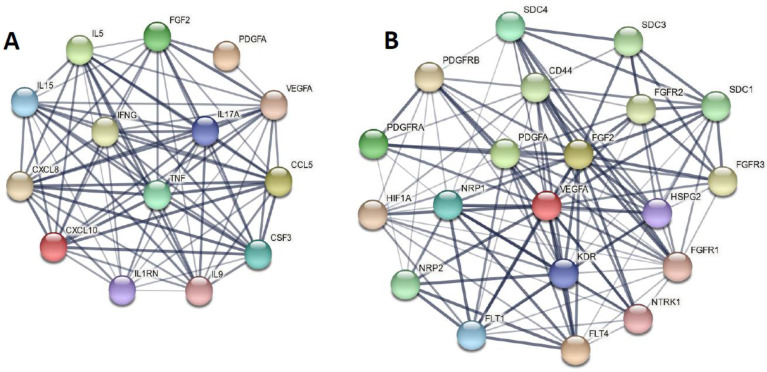
(**A**) The protein–protein interaction network (PPIN) of the 14 differentially expressed immune/growth factors of major dysmood disorder (MDMD). (**B**) The first order PPIN of the growth factor cluster of MDMD.

**Figure 6 cells-11-01183-f006:**
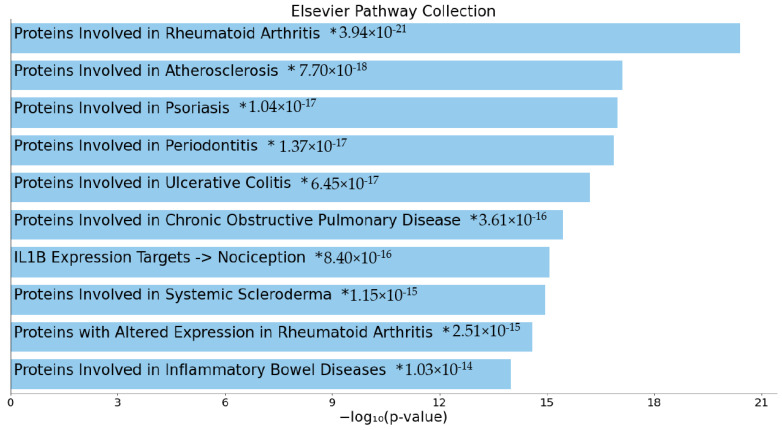
Bar chart with the top ten Elsevier Pathways that are over-represented in the first-order protein–protein interaction network of major dysmood disorder. *: the term has a significant adjusted *p*-value (<0.05).

**Figure 7 cells-11-01183-f007:**
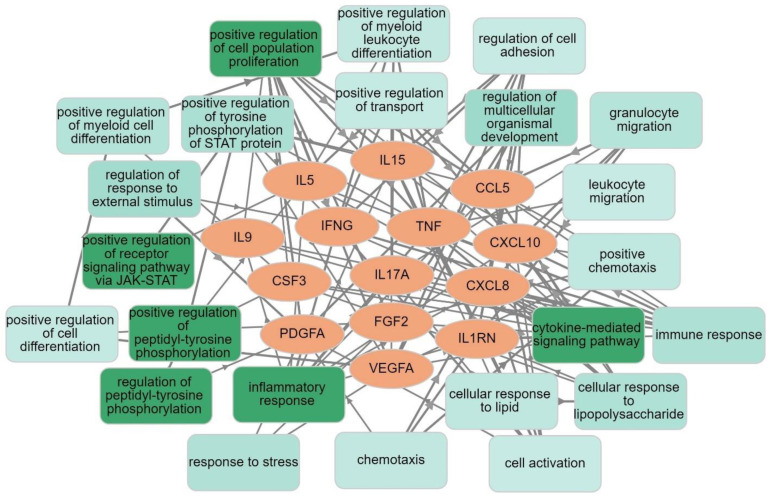
Results of GOnet enrichment analysis showing the 14 seed differentially expressed proteins of major dysmood disorder and their significant GO annotations.

**Table 1 cells-11-01183-t001:** Demographic and clinical data of the healthy controls (HC) and depressed patients divided into those with simple depression and major dysmood disorder (MDMD).

Variables	HC ^a^ (n = 20)	Simple Depression ^b^ (n = 11)	MDMD ^c^ (n = 19)	F/X²/FFHT/KW	df	*p*
Sex (Male/Female)	6/14	4/7	7/12	0.24	2	0.888
Age (years)	33.6 (8.0)	27.0 (5.4)	29.6 (9.9)	2.47	2/47	0.095
Education (years)	16.1 (2.2)	16.5 (0.9)	15.1 (1.3)	2.99	2/47	0.060
BMI (kg/m^2^)	21.33 (2.51)	25.49 (5.55)	25.55 (6.32)	4.32	2/47	0.019
TUD (No/Yes)	18/2	9/2	14/5	1.78	-	0.408
Melancholia-Psychosis (No/Yes)	20/0 ^c^	11/0 ^c^	13/6 ^a,b^	9.14	-	0.003
HDRS	0.9 (1.5) ^b,c^	22.2 (5.7) ^a^	24.3 (5.8) ^a^	147.01	2/47	<0.001
STAI	37.7 (10.6) ^b,c^	56.8 (5.2) ^a^	56.9 (8.2) ^a^	28.00	2/47	<0.001
Number depression episodes	0.0	1.45 (0.52) ^c^	2.31 (1.01) ^b^	KW	-	<0.001
Total number of all episodes	0.0	1.45 (0.52) ^c^	2.47 (0.90) ^b^	KW	-	<0.001
Reoccurrence of illness	−1.084 (0.00) ^b,c^	0.170 (0.353) ^a,c^	1.042 (0.429) ^a,b^	KW	-	<0.001
Lifetime Suicidal behaviors	−0.987 (0.0) ^b,c^	0.044 (0.613) ^a,c^	1.013 (0.611) ^a,b^	KW	-	<0.001
Recent suicidal behaviors	−0.916 (0.0) ^b,c^	0.082 (0.789) ^a,c^	0.917 (0.762) ^a,b^	KW	-	<0.001
LV ROI-immune-growth factors	−1.013 (0.200) ^b,c^	0.034 (0.447) ^a,c^	1.047 (0.522) ^a,b^	127.10	2/47	<0.001
LV Phenome	−1.123 (0.225) ^b,c^	0.504 (0.216) ^a,c^	0.890 (0.500) ^a,b^	170.48	2/47	<0.001

Results are shown as mean ± SD. F: results of analysis of variance; X²: analysis of contingency tables; FFHT: Fisher–Freeman–Halton Exact Test; KW: Kruskal–Wallis test; LV: latent vectors; BMI: body mass index; HDRS: Hamilton Depression Rating Scale score; STAI: Spielberger State and Train Anxiety, State version. ^a,b,c^: pairwise comparisons among sample means.

**Table 2 cells-11-01183-t002:** Differences in unstimulated (UNST) and lipopolysaccharide + phytohemagglutinin-stimulated (STIM) changes in various immune profiles in healthy controls (HC) and depressed patients, divided into those with simple depression and major dysmood disorder (MDMD).

Variables (z Scores)		HC ^a^ (*n* = 20)	Simple Depression ^b^ (*n* = 11)	MDMD ^c^ (*n* = 19)	Wald (df = 2)	*p*
M1	UNST	−0.816 (0.081)	−0.800 (0.108)	−0.947 (0.065)	8.59	0.014
	STIM	0.670 (0.066) ^c^	0.739 (0.126) ^c^	1.136 (0.182) ^a^		
Th1	UNST	−1.438 (0.040)	−1.464 (0.041)	−1.486 (0.024)	9.92	0.007
	STIM	0.168 (0.089) ^c^	0.194 (0.127) ^c^	0.807 (0.204) ^a^		
Th17	UNST	−1.628 (0.047)	−1.739 (0.054)	−1.755 (0.048)	6.44	0.040
	STIM	0.3010 (0.078) ^c^	0.323 (0.123)	0.661 (0.143) ^a^		
Th2	UNST	−1.313 (0.099)	−1.394 (0.118)	−1.289 (0.080)	10.82	0.004
	STIM	0.072 (0.108) ^c^	0.339 (0.228)	0.776 (0.232) ^a^		
IRS	UNST	−1.461 (0.112)	−1.486 (0.178)	−1.617 (0.070)	14.33	0.001
	STIM	0.184 (0.077) ^c^	0.294 (0.169)	0.736 (0.199) ^a^		
CIRS	UNST	−0.870 (0.070)	−0.790 (0.117)	−0.939 (0.076)	8.62	0.013
	STIM	0.718 (0.093) ^c^	0.790 (0.137)	1.100 (0.159) ^a^		
T cell	UNST	−1.408 (0.036)	−1.481 (0.044)	−1.480 (0.035)	14.38	0.001
	STIM	0.095 (0.081) ^c^	0.199 (0.171) ^c^	0.674 (0.144) ^a^		
GF	UNST	−0.816 (0.120)	−0.694 (0.214)	−0.789 (0.117)	13.38	0.001
	STIM	0.506 (0.073) ^c^	0.802 (0.219) c	1.053 (0.206) ^a^		
NT	UNST	−1.603 (0.051)	−1.662 (0.060)	−1.010 (0.032)	12.16	0.002
	STIM	0.277 (0.081) ^c^	0.260 (0.128) ^c^	0.780 (0.144) ^a^		

Results of GEE analyses with immune profiles as dependent variables and time, group (depression versus controls), and time by group interactions as explanatory variables, and age, sex, body mass index, and tobacco use as covariates. Shown are the time × group effects (Wald), with ^a–c^ indicating pairwise comparisons among the study samples. All data are shown as estimated marginal means (mean ± SE). See Appendix A for an explanation of the profiles and cytokines measured in this study. M1: M1 macrophage, Th: T helper, IRS: immune-inflammatory response system, CIRS: compensatory immunoregulatory response system, T cell: T cell growth, GF: growth factors, NT: neurotoxicity.

**Table 3 cells-11-01183-t003:** Differences in lipopolysaccharide + phytohemagglutinin-stimulated changes in cytokines/growth factors in healthy controls (HC) and depressed patients, divided into those with simple depression and major dysmood disorder (MDMD).

Variables (z Scores)	HC ^a^	Simple Depression ^b^	MDMD ^c^	Wald (df = 2)	*p*-Value
sIL-1RA	0.517 (0.088) ^c^	0.399 (0.108) ^c^	1.059 (0.124) ^a,b^	17.17	<0.001
IL-5	−0.2991 (0.079) ^c^	0.100 (0.271)	0.612 (0.309) ^a^	10.62	0.005
CXCL8	−0.100 (0.084) ^c^	0.113 (0.222)	0.852(0.385) ^a^	8.94	0.011
IL-9	−0.055 (0.059) ^c^	0.081 (0.105)	0.498 (0.215) ^a^	9.31	0.010
IL-15	−0.031 (0.102) ^c^	0.409 (0.206)	0.607 (0.144) ^a^	13.13	0.001
IL-17	0.028 (0.081) ^c^	0.249 (0.179)	0.587 (0.209) ^a^	8.59	0.014
FGF	0.277 (0.085) ^c^	0.290 (0.137) ^c^	0.620 (0.069) ^a,b^	10.33	0.006
G-CSF	−0.274 (0.024) ^c^	0.195 (0.295)	0.689 (0.323) ^a^	10.79	0.005
IFN-γ	0.352 (0.119) ^c^	0.437 (0.185) ^c^	0.920 (0.163) ^a,b^	9.32	0.009
CXCL10	0.313 (0.086) ^c^	0.395 (0.116)	0.664 (0.092) ^a^	10.67	0.005
PDGF	−0.321(0.099) ^c^	0.043 (0.298) ^c^	0.415 (0.282) ^a,b^	10.34	0.006
CCL5	0.100 (0.114) ^c^	0.207 (0.132)	0.597 (0.195) ^a^	6.66	0.036
TNF-α	−0.082 (0.092) ^c^	−0.087 (0.150)	0.389 (0.215) ^a^	7.60	0.022
VEGF	0.058 (0.118) ^c^	0.345 (0.171)	0.626 (0.115) ^a^	8.97	0.005

Results of GEE analyses with cytokines/growth factors as dependent variables and time, group (depression versus controls), and time by group interactions as explanatory variables. Shown are the time × group effects (Wald), with ^a–c^ indicating pairwise comparisons among the groups. All data are shown as estimated marginal means (mean ± SE) after covarying for age, sex, smoking, and body mass index.

**Table 4 cells-11-01183-t004:** Associations between immune/growth factor profiles and the features of major dysmood disorder, namely reoccurrence of illness (ROI), Hamilton Depression rating Scale (HDRS) score, and lifetime and recent suicidal behaviors (SB).

Variables	Features	B	SE	W	*p*
M1	ROI	0.186	0.0679	7.23	0.006
	HDRS	0.015	0.0069	4.96	0.026
	PC lifetime SB	0.190	0.0747	6.47	0.011
	PC recent SB	0.423	0.204	4.30	0.038
Th1	ROI	0.226	0.0914	6.09	0.014
	HDRS	0.017	0.0068	6.06	0.014
	PC lifetime SB	0.165	0.0774	4.55	0.033
	PC recent SB	0.612	0.2154	8.08	0.004
Th17	ROI	0.152	0.0674	5.11	0.024
	HDRS	0.010	0.0062	2.76	0.097
	PC lifetime SB	0.136	0.0752	3.28	0.070
	PC recent SB	0.339	0.1872	3.29	0.070
Th2	ROI	0.205	0.1014	4.08	0.043
	HDRS	0.026	0.0082	9.93	0.002
	PC lifetime SB	0.178	0.0963	3.40	0.065
	PC recent SB	0.521	0.2177	8.73	0.017
IRS	ROI	0.247	0.0772	10.20	0.001
	HDRS	0.020	0.0070	7.99	0.005
	PC lifetime SB	0.227	0.0764	8.80	0.003
	PC recent SB	0.532	0.1860	8.18	0.004
CIRS	ROI	0.158	0.0695	5.18	0.023
	HDRS	0.007	0.0074	0.94	0.331
	PC lifetime SB	0.173	0.0728	5.67	0.017
	PC recent SB	0.197	0.1576	1.56	0.212
T cell growth	ROI	0.240	0.0671	12.83	<0.001
	HDRS	0.017	0.0060	7.70	0.006
	PC lifetime SB	0.199	0.0700	7.90	0.004
	PC recent SB	0.366	0.1825	4.02	0.045
Growth factors	ROI	0.199	0.0630	10.04	0.002
	HDRS	0.180	0.0051	12.90	<0.001
	PC lifetime SB	0.173	0.0594	8.45	0.004
	PC recent SB	0.119	0.0651	3.34	0.068
Neurotoxicity	ROI	0.199	0.0790	6.34	0.012
	HDRS	0.130	0.0060	4.80	0.028
	PC lifetime SB	0.209	0.0732	8.15	0.004
	PC recent SB	0.573	0.1670	11.78	0.001

All results of GEE analyses with immune/growth factor profiles as dependent variables and time, feature, and the time × feature interaction as explanatory variables. Shown are the time × feature interactions.

**Table 5 cells-11-01183-t005:** KEGG pathway classifications of the differently expressed proteins (DEPs) in major dysmood disorder.

**Term ID All DEPs**	**Term Description**	**Observed**	**Background**	**Strength**	**FDR**
hsa04060	Cytokine–cytokine receptor interaction	33	282	1.55	5.66 × 10^−40^
hsa04061	Viral protein interaction with cytokine and cytokine receptor	18	96	1.76	9.11 × 10^−24^
hsa05200	Pathways in cancer	27	517	1.2	2.34 × 10^−23^
hsa04657	IL-17 signaling pathway	17	92	1.75	1.56 × 10^−22^
hsa05163	Human cytomegalovirus infection	20	218	1.45	2.33 × 10^−21^
hsa04668	TNF signaling pathway	16	112	1.64	1.15 × 10^−19^
hsa04630	JAK-STAT signaling pathway	17	160	1.51	4.88 × 10^−19^
hsa04010	MAPK signaling pathway	19	288	1.3	7.36 × 10^−18^
hsa05142	Chagas disease	13	99	1.6	1.72 × 10^−15^
hsa04151	PI3K-Akt signaling pathway	18	350	1.2	4.03 × 10^−15^
**Term ID cluster2**	**Term Description**	**Observed**	**Background**	**Strength**	**FDR**
hsa04014	Ras signaling pathway	12	226	1.72	3.05 × 10^−16^
hsa04010	MAPK signaling pathway	12	288	1.61	2.54 × 10^−15^
hsa04015	Rap1 signaling pathway	11	202	1.73	3.35 × 10^−15^
hsa04151	PI3K-Akt signaling pathway	12	350	1.53	1.23 × 10^−14^
hsa01521	EGFR tyrosine kinase inhibitor resistance	8	78	2	8.03 × 10^−13^
hsa05205	Proteoglycans in cancer	9	196	1.65	1.08 × 10^−11^
hsa05200	Pathways in cancer	11	517	1.32	3.19 × 10^−11^
hsa05230	Central carbon metabolism in cancer	7	69	2	3.19 × 10^−11^
hsa04510	Focal adhesion	7	198	1.54	3.20 × 10^−28^
hsa04810	Regulation of actin cytoskeleton	7	209	1.52	4.15 × 10^−8^
**Term ID All DEPs**	**Disease Desription**	**Observed**	**Background**	**Strength**	**FDR**
DOID:2914	Immune system disease	22	611	1.04	1.06 × 10^−13^
DOID:612	Primary immunodeficiency disease	17	426	1.09	8.57 × 10^−11^
DOID:0050589	Inflammatory bowel disease	9	52	1.72	5.12 × 10^−10^
DOID:417	Autoimmune disease	14	294	1.16	1.05 × 10^−9^
DOID:0060180	Colitis	7	22	1.99	3.60 × 10^−9^
DOID:0060032	Autoimmune disease of musculoskeletal system	11	170	1.3	9.45 × 10^−9^
DOID:0050117	Disease by infectious agent	13	317	1.1	2.25 × 10^−8^
DOID:37	Skin disease	15	481	0.98	2.25 × 10^−8^
DOID:77	Gastrointestinal system disease	15	510	0.95	4.38 × 10^−8^
DOID:65	Connective tissue disease	17	715	0.86	4.60 × 10^−8^

KEGG: Kyoto Encyclopedia of Genes and Genomes; ID: Identification; FDR: false discovery rate; IL-17: interleukin; TNF: tumor necrosis factor; JAK-STAT: Janus-kinases–signal transducer and activator of transcription proteins; MAPK: mitogen-activated protein kinase; PI3K-Akt: Phosphatidylinositol 3-kinase–protein kinase B; EGFR: epidermal growth factor receptor.

**Table 6 cells-11-01183-t006:** Results of Molecular Complex Detection (MCODE) analysis performed on differentially expressed proteins (DEPs) of major dysmood disorder.

MCODE Components	GO ID	Biological Term	Log_10_ (*p*) Value
DEPs cluster 1, MCODE1	GO:0019221	Cytokine-mediated signaling pathway	−61.8
	GO:0071345	Cellular response to cytokine stimulus	−46.2
	GO:00.4097	Response to cytokine	−44.5
DEPs cluster 1, MCODE2	GO:0043123	Positive regulation of I-kappaB kinase/NF-kappaB signaling pathway	−6.6
	GO:0043122	Regulation of I-kappaB kinase/NF-kappaB signaling	−6.2
	GO:0019904	Protein domain-specific binding	−4.9
DEPs cluster 1, MCODE3	GO:0097300	Programmed necrotic cell death	−9.3
	GO:0070265	Necrotic cell death	−9.1
	GO:0033209	Tumor necrosis factor-mediated signaling pathway	−8.2
DEPs cluster 2, MCODE	GO:0004714	Transmembrane receptor protein tyrosine kinase activity	−26.0
	GO:0019199	Transmembrane receptor protein kinase activity	−24.6
	GO:0019839	Growth factor binding	−24.5

GO ID: Gene Ontology Identification; ECM: extracellular matrix; TNF: tumor necrosis factor; IL: interleukin. Cluster-1: Immune communality of the MDMD protein–protein interaction network. Cluster-2: Growth factor communality of the MDMD protein–protein interaction network.

**Table 7 cells-11-01183-t007:** REACTOME pathways and PANTHER biological processes statistically over-represented in the differentially expressed proteins of major dysmood disorder.

**REACTOME Pathways**	**Total**	**Expected**	**Hits**	* **p** *	* **pFDR** *
TRIF-mediated TLR3/TLR4 signaling	87	2.14	23	3.35 × 10^−18^	2.08 × 10^−15^
MyD88-independent cascade	88	2.16	23	4.45 × 10^−18^	2.08 × 10^−15^
Toll-Like Receptor 3 (TLR3) Cascade	88	2.16	23	4.45 × 10^−18^	2.08 × 10^−15^
Activated TLR4 signaling	100	2.46	23	9.92 × 10^−17^	3.48 × 10^−14^
Toll-Like Receptor 4 (TLR4) Cascade	103	2.53	23	2.01 × 10^−16^	5.62 × 10^−14^
Innate Immune System	521	12.8	47	7.42 × 10^−16^	1.73 × 10^−13^
Immune System	1140	28	71	1.30 × 10^−15^	2.60 × 10^−13^
Signaling by Interleukins	116	2.85	23	3.25 × 10^−15^	5.69 × 10^−13^
Cytokine Signaling in Immune system	286	7.02	34	5.38 × 10^−15^	8.37 × 10^−13^
Toll-Like Receptor Cascades	123	3.02	23	1.25 × 10^−14^	1.75 × 10^−12^
**PANTHER Biological Processes**	**Total**	**Expected**	**Hits**	** *p* **	** *pFDR* **
Transcription by RNA polymerase II	626	10.4	52	5.34 × 10^−23^	1.04 × 10^−20^
Immune response	387	6.4	33	5.71 × 10^−15^	5.53 × 10^−13^
Viral process	448	7.41	32	2.03 × 10^−12^	1.32 × 10^−10^
Angiogenesis	252	4.17	23	2.61 × 10^−11^	1.27 × 10^−9^
Negative regulation of apoptotic process	577	9.54	33	3.42 × 10^−10^	1.33 × 10^−8^
Transcription, DNA-templated	217	3.59	20	4.69 × 10^−10^	1.52 × 10^−8^
Rhythmic process	124	2.05	15	1.83 × 10^−9^	5.08 × 10^−8^
Apoptotic process	699	11.6	29	4.10 × 10^−6^	9.95 × 10^−5^
Regulation of binding	4	0.0661	3	1.76 × 10^−5^	0.000379
Cell differentiation	971	16.1	34	2.21 × 10^−5^	0.000428

FDR: false discovery rate; TRIF: TIR-domain-containing adapter-inducing interferon-β; TLR: Toll-Like receptor.

## Data Availability

The dataset generated during and/or analyzed during the current study will be available from the corresponding author (M.M.) upon reasonable request and once the dataset has been fully exploited by the authors.

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
