# Peer review of "The Immune Profile of Major Dysmood Disorder: Proof of Concept and Mechanism Using the Precision Nomothetic Psychiatry Approach"

_cells, 2022, doi:10.3390/cells11071183_

Round 1

Reviewer 1 Report

The authors present an interesting study addressed at the characterisation of the involvement of inflammatory parameters in the new diagnostic class namely Major DysMood Disorder (MDMD).

I only have minor suggestions to improve the quality of the ms.

1- The Introduction in quite wordy and partly misleading. In more details,

a) I understand that the authors had key experience in the study of the involvement of inflammatory pathways in psychopathologies, but other studies must also be cited.

b) I see no reason for talking about bipolar disorder in the introduction.

2- I would suggest to avoid the black background in Figure 7, since this is not helping the understanding of the enrichment analysis.

I understand that the authors have key experience in the study of the involvement of inflammatory pathways in psychopathologies, but in the introduction they are citing almost exclusively their own publications (1-4, 6, 8-12).

Author Response

The authors present an interesting study addressed at the characterisation of the involvement of inflammatory parameters in the new diagnostic class namely Major DysMood Disorder (MDMD).

I only have minor suggestions to improve the quality of the ms.

1- The Introduction in quite wordy and partly misleading. In more details,

  1. a) I understand that the authors had key experience in the study of the involvement of inflammatory pathways in psychopathologies, but other studies must also be cited.

@ANSWER: I have added a sentence that all these primary findings were replicated. It reads:

These early findings in MDE are now well replicated in various systematic reviews and meta-analyses which also include the effects of antidepressant treatments on cytokines and inflammation [4-13].

and we added 10 new references.

  1. b) I see no reason for talking about bipolar disorder in the introduction.

@ANSWER: bipolar disorder is deleted (partly) from the Introduction, except when the nomothetic models are described because these include both MDD and BD.

2- I would suggest to avoid the black background in Figure 7, since this is not helping the understanding of the enrichment analysis.

@ANSWER: now changed into the same pic but with a white background.

I understand that the authors have key experience in the study of the involvement of inflammatory pathways in psychopathologies, but in the introduction they are citing almost exclusively their own publications (1-4, 6, 8-12).

@ANSWER: I added 10 other papers.

Reviewer 2 Report

The submitted article by Maes et al. describes activation of cytokines and altered immunological profiles between simple deppression patients and MDMD patients. The experiments have been well conceived and executed along with aproppriate statistical analysis. There are just a couple of minor issues which need to be corrected before this manuscript can be accepted for publication.

Minor issues:

Introduction is well written, and results are clearly presented, nevertheless previous work has not been cited properly, authors should cite more original publications.

Major findings in this work show significantly elevated levels of inflammatory cytokines in the Major Dysmood Disorder depressed patient cohort, which are in line with suicidal behavior and the phenome of depression. Results presented here are in agreement with previous studies, and could serve as a potential biomarkers of depression.

Line 24: Major Dysmood Disorder is not properly abbreviated in the abstract.

Line 181: It is not clear which ammount of LPS has been used in this study 25ug/ml or 25mg/ml?

Author Response

The submitted article by Maes et al. describes activation of cytokines and altered immunological profiles between simple deppression patients and MDMD patients. The experiments have been well conceived and executed along with aproppriate statistical analysis. There are just a couple of minor issues which need to be corrected before this manuscript can be accepted for publication.

Minor issues:

Introduction is well written, and results are clearly presented, nevertheless previous work has not been cited properly, authors should cite more original publications.

@ANSWER: I added 10 more reviews in the Introduction.

Major findings in this work show significantly elevated levels of inflammatory cytokines in the Major Dysmood Disorder depressed patient cohort, which are in line with suicidal behavior and the phenome of depression. Results presented here are in agreement with previous studies, and could serve as a potential biomarkers of depression.

@THANK you.

Line 24: Major Dysmood Disorder is not properly abbreviated in the abstract.

@ANSWER: corrected. See abstract.

Line 181: It is not clear which ammount of LPS has been used in this study 25ug/ml or 25mg/ml?

@ANSWER: adjusted.